

# A new OCO-2 cloud flagging and rapid retrieval of marine boundary layer cloud properties

Mark Richardson[1,2], Matthew D. Lebsock[1], James McDuffie[1], Graeme L. Stephens[1,2,3]

[1]Jet Propulsion Laboratory, California Institute of Technology, Pasadena, CA 91109, USA
[2]Colorado State University, Fort Collins, CO 90095, USA
[3]Department of Meteorology, University of Reading, RG6 7BE, UK

*Correspondence to*: Mark Richardson (markr@jpl.nasa.gov)

**Abstract.** The Orbiting Carbon Observatory-2 (OCO-2) carries a hyperspectral A-band sensor that can obtain information about cloud geometric thickness ($H$). The OCO2CLD-LIDAR-AUX product retrieved $H$ with the aid of collocated
CALIPSO lidar data to identify suitable clouds and provide *a priori* cloud-top pressure ($P_{top}$). This collocation is no longer possible since CALIPSO's coordination flying with OCO-2 has ended, so here we introduce a new cloud flagging and *a priori* assignment using only OCO-2 data, restricted to ocean footprints where solar zenith angle < 45°. Firstly, a multi-layer perceptron network was trained to identify liquid clouds over ocean with sufficient optical depth ($\tau > 1$) for a valid retrieval, and agreement with MODIS-CALIPSO is 90.0 %. Secondly, we developed a lookup table to simultaneously retrieve cloud $\tau$,
effective radius ($r_e$) and $P_{top}$ from A-band and $CO_2$ band radiances, with the intention that these will act as the *a priori* in a future retrieval. Median $P_{top}$ difference versus CALIPSO is 12 hPa with interdecile range [-11,87] hPa, substantially better than the MODIS-CALIPSO [-83,81] hPa. The MODIS-OCO-2 $\tau$ difference is 0.8 (-3.8,6.9) and $r_e$ is -0.3 [-2.8,2.1] μm. The $\tau$ difference is due to optically thick and horizontally heterogeneous cloud scenes. As well as an improved passive $P_{top}$ retrieval, this *a priori* information will allow a purely OCO-2 based Bayesian retrieval of cloud droplet number
concentration ($N_d$). Finally, our cloud flagging procedure may also be useful for future partial column above-cloud $CO_2$ abundance retrievals.

## 1.   Introduction

Hyperspectral O₂ A-band measurements near $\lambda = 0.78$ μm, such as those taken by the Orbiting Carbon Observatory-2 (OCO-2), may provide unique new information about boundary layer clouds by retrieving their geometric thickness ($H$) or droplet number concentration ($N_d$). They are able to do this because the spectrum responds to the photon path length between the Sun, Earth and the sensor. Increased $H$ or decreased $N_d$ with all other cloud properties held constant leads to increased distance between within-cloud scattering events, and therefore a longer photon path length and decreased transmittance in



wavelengths where $O_2$ absorbs. This leads to spectrally varying changes in observed A-band spectra that can allow joint retrievals of cloud optical depth ($\tau$), cloud top pressure ($P_{top}$) and $H$, provided there is sufficient spectral resolution and low enough noise (O'Brien and Mitchell, 1992; Richardson and Stephens, 2018). Equivalently, with knowledge of a relevant effective radius ($r_e$), the $N_d$ could in principle be retrieved along with $\tau$ and $P_{top}$.

The basic principle of A-band absorption for cloud height is well established (Fischer and Grassl, 1991; Rozanov and

Kokhanovsky, 2004; Yamamoto and Wark, 1961) and numerous spaceborne A-band instruments retrieve cloud properties (Koelemeijer et al., 2001; Kokhanovsky et al., 2005; Lindstrot et al., 2006; Loyola et al., 2018; Preusker et al., 2007; Vanbauce et al., 1998), but most lack the spectral resolution or noise characteristics to obtain $H$ (e.g. Schuessler et al. (2014)). Others rely on multi-angle (Ferlay et al., 2010) or combined A- and B-band information (Yang et al., 2013), although these tend to contain little information on low-altitude and relatively thin clouds like marine stratocumulus (Davis

et al., 2018; Merlin et al., 2016).

An OCO-2 based retrieval of $\tau$, $P_{top}$ and $H$ has been developed (OCO2CLD-LIDAR-AUX, available at www.cloudsat.cira.colostate.edu/data-products/level-aux/oco2cld-lidar-aux), which uses lidar-based retrievals from the Cloud-Aerosol Lidar and Infrared Pathfinder Satellite Observation (CALIPSO) satellite to help identify cloudy scenes and constrain prior $P_{top}$ (Richardson et al., 2019). This retrieval is targeted at single-layer liquid clouds over the ocean whose

response, both to warming and aerosols, are a major source of uncertainty in climate simulations (e.g. Bony et al. (2005); Bodas-Salcedo et al., (2019); Zelinka et al. (2020)). Independent information about cloud structure may help to address timely questions where other sensors which rely on different retrieval approaches and assumptions can lead to apparently contradictory conclusions (Rosenfeld et al., 2019; Toll et al., 2019).

With CALIPSO leaving the A-Train constellation in 2018, collocation between OCO-2 and CALIPSO footprints is no longer

possible. Our future retrievals require a new cloud flagging method plus *a priori* cloud top information for our iterative Bayesian optimal estimation (OE) retrieval (Rodgers, 2000). This paper describes a new pre-processor for OCO-2 based liquid cloud property retrievals that provides the requisite cloud flagging and *a priori* information. Details of OCO2CLD-LIDAR-AUX are summarised in Table 1, which also lists the main changes introduced in this study.

We do not use the published OCO-2 cloud flag as it was not developed for ocean nadir scenes (Taylor et al., 2016), since

they were considered too dark for OCO-2's main mission of column $CO_2$ (XCO2) retrievals (Crisp, 2008; Crisp et al., 2004; Eldering et al., 2016). Therefore we train a multi-layer perceptron network to rapidly identify liquid cloud scenes using collocated CALIPSO and Moderate Resolution Imaging Spectroradiometer (MODIS) retrievals. For the prior cloud property retrieval we develop lookup tables (LUTs) that jointly retrieve $\tau$, $r_e$ and $P_{top}$ using OCO-2 $O_2$ A-band and strong $CO_2$ band ($\lambda$ ~ 2.06 μm) radiances. These are similar to the Nakajima-King tables used in MODIS cloud retrievals (Nakajima and King,

1990) but add an A-band absorption ratio that is sensitive to $P_{top}$.

Our OCO-2 OE retrievals are computationally expensive due to the complex radiative transfer (RT), so we aim to avoid footprints which are unlikely to yield good retrievals. The cloud flagging and prior LUT retrieval developed here are a necessary step in excluding these footprints, and we further exclude those where solar zenith angle, SZA > 45° based on



OCO2CLD-LIDAR-AUX's retrieval statistics. It is possible that a future partial-column (i.e. above cloud) XCO2 retrieval
could be developed, which would likely be targeted at columns above optically thick clouds, so the pre-processor developed
here could find wider use (Schepers et al., 2016; Vidot et al., 2009). A further development is that our past retrievals used a
fixed $r_e$ and the addition of varying $r_e$ is eased by a new Python RT interface using the ReFRACtor (Reusable Framework for
the Retrieval of Atmospheric Composition) software described in Section 2.3. Our new LUT retrieval of a prior $r_e$ will allow
a more appropriate $r_e$ to be assumed in the iterative OE.

The paper is organised as follows: Section 2 describes the relevant OCO-2 details, data selection and radiative transfer
calculations before detailing the methodology. Section 3 reports the performance statistics of the classifier, compares LUT
retrieved cloud properties versus MODIS and CALIPSO where the instrument footprints overlap, and compares the final
pre-processor throughput against that of OCO2CLD-LIDAR-AUX. Section 4 discusses and contextualises the results as well
as proposing actionable future work that could address identified biases and Section 5 concludes.

## 2.  Methods and data

### 2.1. Instruments and data selection

The OCO-2 measurement approach and instrumentation are detailed in Bösch et al. (2017), the L2FP RT's application to
clouds in Richardson et al. (2017), and the MODIS-CALIPSO-OCO-2 matchup data are as used in Taylor et al. (2016). The
datasets used here are listed in Table 2, in particular, from the OCO-2 Level 1b Science (L1bSc) data we obtain calibrated
radiances and RT inputs such as solar zenith angle (SZA) and instrument characteristics.

The OCO-2 satellite flies in the Sun-synchronous A-Train constellation (L'Ecuyer and Jiang, 2010), and measures during the
daytime ascending node with an equator crossing time near 1:30 pm. Its orbits are committed primarily to either glint or
nadir view, and we use nadir only orbits. It carries three co-boresighted grating spectrometers centred over the $O_2$ A-band
($\lambda{\sim}0.78$ µm), weak $CO_2$ band ($\lambda{\sim}1.68$ µm) and strong $CO_2$ band ($\lambda{\sim}2.06$ µm).

The satellite operates in a pushbroom fashion with a swath of 8 footprints whose orientation relative to the track rotates
through the orbit as the satellite angles to optimise solar power generation. The subsequent parallelogram-like footprints are
nominally near 1.4 km×2.2 km at nadir. The channels' wavelengths vary across the track due to the manner in which the
optics focus light onto the focal plane array (FPA), and wavelength also drifts throughout an orbit due to Doppler shift. This
causes issues for a LUT developed from a fixed set of channels, since the wavelengths sampled by those channels will differ
between each measurement. Furthermore, some sensor pixels are damaged and we only include channel indexes where all 8
swath soundings are classed as good, which reduces the A-band sample from 1,016 to 853 channels. Section 2.2 describes
how we use a channel averaging approach to reduce the consequences of this wavelength shift in the cloud classifier and
Section 2.4 our related channel selection for the LUT.

For classifier training and validation, we require spatial overlap between OCO-2, MODIS and CALIPSO data. The
ascending OCO-2 ground track is approximately 200 km to the east of Aqua's and therefore within the MODIS swath, so we



select the 1 km MODIS retrieval footprint whose centre is closest at the surface to the centre of the OCO-2 footprint. However, CALIPSO only measures once at nadir so only one OCO-2 footprint in each swath can be collocated. Furthermore, even during formation flying the satellites drifted within their control boxes and some CALIPSO measurements occurred outside the OCO-2 swath. We only include footprints with a CALIPSO-OCO-2 matchup distance of

<1.5 km at the surface. Finally, the dataset was further restricted to footprints with surface type = water, SZA < 45° and with valid radiances. Between 2014-09-06 and 2018-04-30 the MODIS-CALIPSO-OCO-2 matchup dataset has 5,909 nadir orbits of which 4,743 contain valid matchups. This is reduced to $N = 3,907$ orbits through 2016-12-31 when we also require an OCO2CLD-LIDAR-AUX retrieval.

## 2.2. Cloud Classifier Data Selection and Training

For the first step of rapidly identifying footprints that contain liquid clouds over the ocean we select a machine learning classifier which is trained on a set of collocated MODIS-CALIPSO footprints before being validated against an independent set of MODIS-CALIPSO data. The footprints which pass this classifier will be forwarded to the LUT estimator to generate the *a priori* cloud property estimate.

We generated independent sets of training ($N$=100,000) and validation ($N$=250,000) footprints by randomly selecting orbits
and taking all their valid footprints until we had those sample sizes. We assign a cloud flag value of 1 to a footprint when the following conditions are all met, else it is 0:

    (i)        CALIPSO Feature_Classification_Flag = 2 (cloud present and is liquid)

    (ii)      CALIPSO retrieves a single layer

    (iii)     MODIS Cloud_Optical_Thickness > 1 (cloud present and is sufficiently optically thick)

As input we take the continuum radiances ($I_c$) from all 3 OCO-2 bands and correct for illumination geometry via $\mu_0^{-1}I_c$ where $\mu_0$=cos(SZA), plus a number of A-band ratios described below. From Python's sklearn package we selected a multi-layer perceptron network (sklearn.neural_networks.MLPClassifier) with hidden layer sizes of (100,50,25). These selections are justified in Supplementary §1.

In these bands the ocean is dark and reflectance increases monotonically with $\tau$, so the $\mu_0^{-1}I_c$ help to identify optically thick
clouds. Ice also absorbs more strongly than water in the higher wavelength bands, which aids phase discrimination.

We calculate A-band absorption ratios by dividing a non-continuum (i.e. absorbing) channel radiance $I_{abs,O2}$ by $I_{c,O2}$. Clouds tend to increase $I_{abs,O2}/I_{c,O2}$ since photons scattered from the clouds encounter fewer $O_2$ molecules than those that travel all the way to the surface. This principle has been exploited to improve detection of clouds over bright snow & ice surfaces with the A- and B-band channels of the Earth Polychromatic Imaging Camera (EPIC) on board the Deep Space Climate
Observatory (DISCOVR) (Zhou et al., 2020). Consider:

$$\frac{I_{abs,O2}}{I_{c,O2}} = \exp\left(-\int k_{O2}(z)\, dz\right), \tag{1}$$



Where $k_{O2}(z)$ is the O₂ absorption coefficient, which is then integrated over the photon path $\int dz$. Considering a δ-function distribution of photon path lengths along the beam that is scattered from a single layer with constant $k_{O2}(z) = k_{O2}$, then at nadir the path can be decomposed into the path from TOA to the layer top, $\mu_0^{-1}\Delta z$, and from the layer top to TOA $\Delta z$:

$$\frac{I_{abs,O2}}{I_{c,O2}} = \exp(-k_{O2}(\mu_0^{-1} + 1)\Delta z), \tag{2}$$

making $k\Delta z$ the subject:

$$k\Delta z = -\ln\left(\frac{I_{abs,O2}}{I_{c,O2}}\right)(\mu_0^{-1} + 1)^{-1}, \tag{3}$$

We take the right-hand side of Eq. (3) as our observable. If we select channel combinations with near-constant $k_{O2}$, then the observable is proportional to $\Delta z$. Lower values should be associated with high (i.e. more likely ice clouds), and high values
with clear scenes. This assumes similar scattering properties for the $I_{abs,O2}$ and $I_{c,O2}$, which is justified by the A-band's small wavelength range.

The $k_{O2}$ sampled by individual channels varies for three main reasons:

(i) The central wavelength of each channel depends on the cross-track position due to the way in which the optics focus light on the FPA,

(ii) The wavelengths sampled change due to Doppler shift induces by relative Earth-satellite and Earth-Sun motion,

(iii) The strength of O₂ absorption varies due to line broadening induced by atmospheric conditions.

We use a method from Richardson et al. (2017) to address these factors. The 853 undamaged channels are ranked from brightest to darkest and a non-overlapping 10-channel mean is taken, resulting in 85 full "super channels". These are combined with $I_{c,O2}$ and $\mu_0$ using Equation (3), and we selected every 10[th] super channel from the 35[th] onwards
(Supplementary §1 shows little improvement from additional super channels).

This is illustrated in Figure 1, with Figure 1(a) showing an example cloudy spectrum and the damaged channels, Figure 1(b) the ranked super channels and those used in the classifier, and Figure 1(c) compares $I_{abs,O2}/I_{c,O2}$ for the original spectrum (CALIPSO $P_{top}$ = 827 hPa) and for a spectrum with similar $\mu_0$ and $I_{c,O2}$ but with CALIPSO $P_{top}$ = 403 hPa. The brightest super channels show little response to scattering layer altitude, so they contain little information and they are excluded from
the classifier. The higher altitude cloud has brighter $I_{abs,O2}$ due to the shorter mean path length. As stated previously, this aids in the phase classification, and also to discriminating between cloudy and clear scenes since very low $I_{abs,O2}/I_{c,O2}$ is more likely associated with photons scattered from the surface.

### 2.3. Radiative Transfer Simulations and ReFRACtor Interface

The forward RT simulations used to generate the LUT are performed with the ReFRACtor RT code, which is based on
LIDORT with a polarisation correction for low orders of scattering (Natraj and Spurr, 2007; Spurr, 2006). This code makes a semi-infinite plane-parallel assumption, with a correction to the direct beam to account for Earth's sphericity. Angular output is calculated for a handful of wavelengths with 8 streams, with the rest of the spectrum interpolated using a single stream





using the method of O'Dell (2010), which reliably reproduces the higher-stream output. This number of streams was found to reproduce cloudy scene radiances given MODIS and CALIPSO cloud properties (Richardson et al., 2017) and also

matches the selection in Vidot et al. (2009).

OCO2CLD-LIDAR-AUX used OCO-2 L2FP RT and required input L1bSc and meteorology files plus a file containing pressure level and cloud information. Each footprint's output was saved to a file for every OE iteration, adding to a read-write bottleneck. Further inefficiency arose as if any footprint in an orbit included a type of scatterer (e.g. water clouds with $r_e = 10$ μm, which we term wc_010), its scattering properties had to be assigned for every profile in the orbit. For example, if

one footprint contained a wc_010 cloud, every other footprint in the orbit that didn't contain a wc_010 cloud would need an assigned wc_010 profile with extinction = 0.

Here we use a new ReFRACtor, which handles footprints as individual objects. Inputs are assigned uniquely to that object and it stores the RT output and updated properties internally, so no external reading or writing is required for intermediate OE iterations.

For LUT input we take an ocean footprint near 25 °S from the L1bSc file for orbit 16094a on 2017-07-11 for instrument and satellite properties, although we manually vary SZA. We used the mean OCO-2 cloudy profiles for tropical (20 °S—20 °N) footprints from Richardson and Stephens (2018). The high latitude case is excluded as its surface temperature is near 0 °C, so will mostly represent ice and mixed-phase clouds, and using the midlatitude (20—50 °S/N) case had little effect on the retrieval statistics.

The RT code takes input on levels and then linearly interpolates to generate vertically homogeneous layers. We use 16 pressure levels: 3 assigned linearly in $P$ from TOA to 500 hPa, then 10 from 500 hPa to $P_{top}$, 2 from $P_{top}$ to cloud bottom ($P_{bottom}$), and the final level is the surface. This was found to reliably reproduce OCO-2 L2FP RT standard outputs which use 20 levels, but with faster processing.

The cloud extinction is assigned to the level at the cloud centre, whose neighbouring levels are at $P_{top}$ and $P_{bottom}$ and the

layer interpolation results in a vertically homogeneous cloud with constant $\tau(z)$ and $r_e(z)$. Rozanov and Kokhanovsky (2004) showed that a vertically uniform assumption may introduce radiation biases, relative to our target marine boundary layer clouds which tend to be vertically stratified with increasing extinction towards cloud top (Bennartz, 2007; Grosvenor et al., 2018; Painemal and Zuidema, 2011), but quantifying such a bias requires extensive testing that we intend to perform separately. For now, the cloud $H$ is calculated as in Szczodrak et al. (2001) where $H \propto \sqrt{\tau r_e}$, and is converted to $\Delta P$ by

assuming $\Delta z / \Delta P \approx 10$ m hPa$^{-1}$. Where this would result in $P_{bottom} > P_{surf}$, the cloud is compressed while maintaining the same $P_{top}$.

For surface reflection ReFRACtor does not currently allow for a Cox-Munk surface, so we assume Lambertian with albedo that varies by band and SZA. The band- and SZA-dependent values are derived from a set of OCO-2 radiances as described in Supplementary §2 and range from 0.010—0.054.


Gaseous absorption is from the absorption coefficient (ABSCO) version 5.0 tables used in OCO-2's latest XCO2 retrieval, version 9. These tables account for line changes due to temperature, pressure and water vapour. Cloud properties are pre-calculated using Mie theory at integer micron values of $r_e$ following an assumed Gamma droplet size distribution with width parameter $\gamma = 1/9$. This follows the standard OCO-2 XCO2 retrieval aerosol input file, but with an update to correct an error in water absorption in the $CO_2$ bands [Aronne Merrelli, pers. comm.].

**2.4. Lookup table development and retrieval**

The LUT is designed to produce prior cloud property estimates for our future OE retrieval, which specifically targets marine boundary layer clouds and aims to provide additional information about their $H$ or $N_d$. We therefore limit the range of the LUT properties to cover the majority of these clouds, with properties $\tau$ from 1—50, $r_e$ from 4—32 μm and $P_{top}$ from 650—970 hPa, and SZA spans 20—45° inclusive (see Supplementary Table 3 for selected values). The simulated outputs are $I_{c,O2}$

in the $O_2$ A-band, $I_{c,st}$ in the strong $CO_2$ band and an A-band ratio $I_{abs,O2}/I_{c,O2}$.

We take the mean of 5 channels for each of $I_{c,O2}$ and $I_{c,st}$ and 10 channels for $I_{abs,O2}$, and fixed channel indices are required to consistently convert the RT simulated spectra into LUT radiances. The selected channels minimise the root mean squared error (RMSE) across a large sample of footprints against the L1bSc continua (for $I_{c,O2}$ and $I_{c,st}$) and the 60th super-channel for $I_{abs,O2}$ (as defined in Section 2.2). The 60th super channel is picked as it showed the greatest sensitivity to CALIPSO $P_{top}$ in

Richardson et al. (2017). The selection algorithm is described in Supplementary §4, the error statistics are in Supplementary Table 4 and the channel indices in Supplementary Table 5. The error statistics show that our selection is valid for a range of meteorological conditions, illumination geometries, Doppler shifts and for all 8 cross-track sounding positions.

The LUT channels are highlighted in Figure 2, which shows mean spectra from a large sample of cloudy footprints. The channel means with 2σ ranges are shown as shaded bands and are compared with the truth as solid lines. The truth for $I_c$ is

the mean of the sample L1bSc radiance continua, which represent the brightest channels in each footprint and whose channel indices may change with footprint while the $I_{abs,O2}$ truth is the spectrum's 60th super channel. The estimators are consistent with the truth in each case, with the best agreement for $I_{c,O2}$ and a negative bias in $I_{c,st}$. We found that scaling the L1bSc $I_{c,st}$ value by 0.9804 resulted in similar error statistics to using our selected channels, so we use scaled L1bSc $I_{c,st}$ in our LUT retrieval since it those radiances are already loaded for the classifier. The individual $I_{abs,O2}$ channels show a large spread, but

the channel selection algorithm accounts for anti-correlation in their radiances such that the 10-channel mean is consistent with the 60th super channel across all test footprints.

For each SZA, Refractor is run for all combinations of input cloud properties to generate A-band and strong $CO_2$ band radiances for these selected LUT channels. A LUT is generated at each 5° in SZA from 20—45° inclusive, and the retrieval works as follows:

(i)      From an L1bSc file, load the SZA plus radiances to get $I_{abs,O2}$, $I_{c,O2}$ and $I_{c,st}$

(ii)      Convert these into the LUT observables $I_{c,O2}$, $I_{c,st}/I_{c,O2}$, and $I_{abs,O2}/I_{c,O2}$





(iii)    Scale observables onto the nearest LUT SZA using the appropriate μ-related scaling,

(iv)    Interpolate within the LUT to simultaneously estimate $\tau$, $r_e$ and $P_{top}$.

If the observed radiances are outside the LUT values then a NaN is returned and the footprint is flagged as not retrievable.

The footprint is also flagged as likely to contain ice if L2Met $T(P_{top,retrieved}) < 0$ °C. We refer to NaN or $T_{top} < 0$ °C outputs as not being passed by the LUT, since these footprints will not be attempted in our future OE retrieval.

### 2.5. Pre-processor prior validation

The pre-processor is run on the 3,907 orbits used in OCO2CLD-LIDAR-AUX from September 2014—December 2016 where the new L1bSc and L2Met Version 8 files are available along with the collocated MODIS and CALIPSO files, and

where there are any ocean footprints with SZA < 45°. For validation of the LUT we consider only those footprints where the CALIPSO matchup distance < 1.5 km as in Section 2.1, where the MODIS, CALIPSO and LUT retrievals are within the valid LUT property range and where derived OCO-2 $T_{top} > 0$ °C ($N = 1,264,449$). The primary analysis is in the pairwise differences between the LUT retrieved properties and MODIS $\tau$ or $r_e$, and CALIPSO $P_{top}$. The MODIS $P_{top}$ is also evaluated against that of CALIPSO.

### 2.6. Comparison with OCO2CLD-LIDAR-AUX pre-processor

The OCO2CLD-LIDAR-AUX matchups are separated into three sets: those that are not flagged by the classifier, those that are flatted but do not pass the LUT retrieval (due to out-of-range cloud properties or implied $T_{top} < 0$ °C), and those that fully pass the pre-processor. Throughput and agreement are compared with the OCO2CLD-LIDAR-AUX cloud flag, retrieval $\chi^2$, retrieved $\tau$ and $P_{top}$ discrepancy versus CALIPSO. The pre-processor performs well if it successfully passes those footprints

with small posterior $\chi^2$ and $P_{top}$ discrepancy while avoiding those with larger values.

The OCO2CLD-LIDAR-AUX cloud flag was based on simple thresholds in $\mu_0^{-1}I_{c,O2}$ and $\mu_0^{-1}I_{c,wk}$ combined with a phase discrimination based on their combination, and finally a requirement for valid CALIPSO single-layer clouds with $P_{top} > 680$ hPa occurring within 10 km. This flag did not have a SZA cutoff at 45°, so we will also specifically consider comparisons between the outputs of the two pre-processors where SZA < 45°.

### 3.    Results

### 3.1. Cloud classifier test statistics

As in Section 2.2 the classifier output is 1 when we expect a single layer liquid cloud with $\tau > 1$ and 0 otherwise and the validation data, which we also term "truth", is the MODIS-CALIPSO classification. We use the following terms:

(i)    True Positive (TP), classifier = 1, truth = 1

(ii)    False Positive (FP), classifier = 1, truth = 0





(iii)     False Negative (FN), classifier = 0, truth = 1

(iv)     True Negative (TN), classifier = 0, truth = 0

Which are normalised such that TP + FP + FN + TN = 100 %. These can be summarised in a confusion matrix, as is done in Figure 3(a) for the $N$ = 250,000 non-training sample. Its trace is the accuracy score of 90.0 % and the off-diagonal elements

represent potential misclassifications. Figure 3(b) shows that the FNs are largely clouds of lower MODIS $\tau$ than those identified by the classifier, with 29.4 % of FNs having MODIS $\tau$ < 3, compared with 7.3 % of TPs.

Some of these "missed" clouds may be due to collocation error, for example a cloud may average $\tau$ > 1 over the 1 km MODIS footprint, bur not over the larger OCO-2 footprint. The classifier will also have errors: it maximises the accuracy score, and detecting lower $\tau$ clouds may require passing darker scenes which could increase the prevalence of FPs.

Figure 3(c) shows the distribution of CALIPSO $T_{top}$ where retrieved, and shows far more cold-topped clouds in the FP case compared with the TPs, although there is also a $T_{top}$ < 0 °C peak in the FN case. This suggests that the classifier misidentifies some ice clouds as liquid, and also that some of the FNs may in reality be mixed phase clouds that CALIPSO has nevertheless identified as liquid. For example, 24.6 % of FNs have $T_{top}$ < -10 °C, compared with 7.6 % of the TP sample.

Among the false positives, we expected that there would be a larger occurrence of broken or multi-layered clouds, where

thick broken clouds were sufficiently bright to trigger detection or where overlying thin ice clouds have too little effect on the radiances to be flagged as ice. We describe a scene as broken when the MODIS partially cloudy retrieval exists (Cloud_Optical_Thickness_PCL > 0) and a scene as multi-layered when CALIPSO retrieves more than 1 cloud layer, although strictly this can only detect multiple layers when the upper layer does not fully attenuate the lidar. While 11.3 % of the full sample is multi-layered, 40.1 % of the FP cases are, and while 12.2 % of scenes are partially cloudy, 30.4 % of FP

footprints are. Overall, 69.4 % of FPs are associated with multi-layer or broken clouds, or both.

### 3.2. Lookup table matchup performance

Figure 4(a) shows simulated $I_{c,O2}$ and $I_{c,st}$ at SZA = 30° and $P_{top}$ = 810 hPa for all $\tau$ and $r_e$ while Figure 4(b) contains the median and spread of $I_{abs,O2}/I_{c,O2}$ at each fixed $P_{top}$. Most of the $I_{abs,O2}/I_{c,O2}$ variance is explained by $P_{top}$, with spread largely due to changes in within cloud scattering. For example, for $P_{top}$ = 970 hPa, the optically thickest clouds were artificially

compressed to prevent them from extending below the surface thereby reducing the in-cloud path and increasing the maximum $I_{abs,O2}/I_{c,O2}$.

The OCO-2 LUT retrievals are compared with those of MODIS and CALIPSO in Figure 5(a—c) and the MODIS and OCO-2 $P_{top}$ differences relative to CALIPSO are in Figure 5(d). We consider the OCO-2 value minus the other product's value, and report median [10[th], 90[th] percentiles] instead of standard deviation as these distributions are commonly non-Gaussian.

There is good correlation between OCO-2 and other products, with a $\tau$ difference of 0.77 [-3.77,6.93], an $r_e$ difference of -0.25 [-2.78,2.13] μm and a $P_{top}$ difference of 12 [-11,87] hPa. As can be seen in Figure 5(d), the LUT $P_{top}$ retrieval



outperforms that of MODIS, whose difference relative to CALIPSO is -17 [-83,81] hPa, i.e. the OCO-2 interdecile range is approximately 40 % smaller than that of MODIS.

We also divide the $\tau$ and $r_e$ differences by the MODIS reported uncertainty ($\sigma_{\tau,MODIS}$, $\sigma_{re,MODIS}$). If the OCO-2 and MODIS

retrievals were independent Gaussian with equal variance then the standard deviation of OCO-2-MODIS differences would be $\sqrt{2} \approx 1.41\,\sigma_{MODIS}$. We find values of $1.26\,\sigma_{\tau,MODIS}$ and $0.37\,\sigma_{re,MODIS}$, indicating that the $r_e$ retrievals are not independent and that our differences are within the MODIS-reported uncertainties.

We acknowledge discrepancies in the median retrieved $\tau$ and $r_e$, and refer to these as biases. The $\tau$ bias grows both with OCO-2 retrieved $\tau$ and with the horizontal variability of the scene as displayed in Figure 6. For this figure, the samples were

split into deciles according to the LUT retrieved $\tau$ or the MODIS sub-pixel index at $\lambda = 0.66$ μm, which is the standard deviation of the 250 m footprint radiances with a 1 km cloud retrieval, divided by the mean of those radiances. Spatial variability and greater optical depths appear to drive much of the $\tau$ bias but we could not identify a dominant factor consistently correlated with the small $r_e$ bias. These issues are further discussed in Section 4.2.

### 3.3. Pre-processor throughput


The multi-layer perceptron classifier passes 5.5 % of all OCO-2 footprints as $\tau > 1$ liquid clouds, of which 0.9 % return invalid cloud properties from the LUT and a further 0.8 % have implied $T_{top} < 0$ °C, resulting in a final throughput of 3.8 %. This is smaller than OCO2CLD-LIDAR-AUX, which attempted to retrieve 14.1 % of all soundings. However, most of the difference is due to SZA, and when we restrict the denominator to all footprints with SZA < 45° the throughputs are 13.1 %

for OCO2CLD-LIDAR-AUX and 11.7 % for the new classifier, or 8.1 % after the LUT thresholds.
Figure 7 displays histograms of selected OCO2CLD-LIDAR-AUX outputs for SZA < 45° retrievals split into footprints where the new pre-processor passes the footprint (blue), where the LUT returns invalid properties or $T_{top} < 0$ °C (orange), or the classifier does not pass the footprint (green). The new classifier identifies "better" retrievals that ended with smaller fit errors: median $\chi^2 = 7.2\times10^{-7}$, versus $1.3\times10^{-4}$ for those not passed (among those with SZA < 45°). The LUT filtering further

improves the statistics, with median $\chi^2 = 9.8\times10^{-7}$ for those not passed by the LUT retrieval versus $\chi^2 = 6.6\times10^{-7}$ for those successfully retrieved with $T_{top} > 0$ °C. The perceptron network also tends to pass clouds that are more optically thick (median $\tau = 8.6$ vs. 2.4) and to show smaller spread in the difference between OCO-2 and CALIPSO $P_{top}$ (standard deviation of differences, $\sigma = 33$ hPa vs. 55 hPa).

The OCO2CLD-LIDAR-AUX footprints that are excluded by the new pre-processor are consistent with optically thinner

clouds and with poorer quality retrievals. Among the footprints that are passed by the new pre-processor, 17.1 % were not attempted in OCO2CLD-LIDAR-AUX.



## 4. Discussion

### 4.1. Cloud classifier and pre-processor throughput

The cloud classifier's agreement of 90.0 % with MODIS-CALIPSO is similar in performance to the original OCO-2
operational cloud flagging for ocean glint used in the L2FP XCO2 retrieval (Taylor et al., 2016). Furthermore, the multi-
layer perceptron network is lightweight (size <250 kB) and fast. It throughputs 11—13 % of ocean soundings where SZA <
45°, of which under a quarter are poor retrieval candidates according to MODIS-CALIPSO. These cases are consistently
(~69 %) broken or multi-layered cloud scenes, while the missed MODIS-CALIPSO cloud scenes are commonly optically
thinner (~4 times likely to be $\tau$ < 3) and colder (~3 times likelier to have $T_{top}$ < -10 °C) than the hit cloud scenes. These
thinner and colder samples are also likely to be poor candidates for our target future retrieval of droplet number
concentration in warm topped clouds.

Applying the LUT retrieval further reduces the number of footprints that are taken to be liquid clouds with $\tau$ > 1. The
OCO2CLD-LIDAR-AUX retrieval attempted 13.1 % of SZA < 45° footprints, the new classifier-LUT pre-processor passes
8.1 %. Figure 7 showed that the excluded footprints tended to be more optically thin, have larger discrepancy in retrieved
$P_{top}$ relative to CALIPSO, and to have higher $\chi^2$. This suggests that the new pre-processor will pass better retrieval
candidates to the OE code, thereby improving efficiency. Of those that are now passed, 17 % were not passed by
OCO2CLD-LIDAR-AUX. These likely include cases of mis-identification that will result in poor quality retrievals, but may
also include true cloud cases that were not identified in OCO2CLD-LIDAR-AUX. For example, retrievals were previously
classified using the nearest CALIPSO footprint up to 10 km away, and if a cloud was in the OCO-2 field of view but not the
CALIPSO field of view, it would not previously have been passed. Overall, the new pre-processor shows good performance
in terms of identifying scenes which likely contain liquid clouds with sufficient $\tau$.

### 4.2. Lookup table cloud property retrieval

The LUT retrieval shows good correlation with MODIS $\tau$ and $r_e$ plus CALIPSO $P_{top}$ in Figure 5. Versus CALIPSO, the
LUT-based $P_{top}$ retrievals have a smaller-magnitude bias and 40 % smaller inter-decile range than MODIS. The 12 hPa $P_{top}$
bias represents OCO-2 retrieved clouds that are lower in the atmosphere than retrieved by CALIPSO. These statistics may
include cases of broken cloud, either above a lower cloud or above the surface. Three-dimensional (3d) cloud effects, or
combined scattering from multiple cloud layers could lead to longer mean photon path lengths and thereby a larger OCO-2
$P_{top}$, assuming that CALIPSO tends to identify the highest layer. We consider full 3d radiative transfer treatments to be
beyond the scope of this study but point readers to a wide literature on this topic (Davis and Knyazikhin, 2005; Heidinger
and Stephens, 2002; Kokhanovsky et al., 2007; Várnai and Marshak, 2002).

Aerosol is ignored in these simulations, as previous analysis using CALIPSO aerosol products showed no change in
OCO2CLD-LIDAR-AUX $P_{top}$ bias in response to CALIPSO-identified aerosol (Richardson et al., 2019). Furthermore,



above-cloud scattering aerosol would tend to reduce photon path length and therefore have an opposite effect on $P_{top}$ to our observed bias.

Retrieved $P_{top}$ could also change due to the assumed cloud vertical structure and meteorological profile used in the LUT development. If the cloud vertical structure used in the RT differs from reality, then this could lead to incorrectly simulated within-cloud photon paths. Firstly, if the simulated cloud is too geometrically thin (low $H$) for a given $\tau$, $r_e$ then the within-cloud path length will be too small and the above-cloud path must be lengthened to compensate, resulting in a positive $P_{top}$ bias, and vice versa for too-high simulated $H$. This study improves on the OCO2CLD-LIDAR-AUX prior realistically

varying $H$ with $r_e$ in addition to $\tau$, but a bias may remain. In particular, shallow marine clouds tend to have extinction weighted towards the top which affects the exiting radiance and may introduce $P_{top}$ biases which vary with geometry and cloud properties. We intend to perform a separate and more detailed analysis of how realistic vertical cloud profiles affect simulated OCO-2 radiances, and determine how to account for such a vertical-structure bias.

With regards to meteorology, a warmer and moister profile broadens the $O_2$ absorption lines and we expect stronger resultant

absorption in the selected $I_{abs,O_2}$ channels. Our tropical meteorology may lead to too-strong absorption in non-tropical scenes such that the retrieved cloud is lifted (i.e. lower $P_{top}$) to compensate, but the observed bias is opposite to this. We also retrieved using a LUT developed with the Richardson and Stephens (2018) midlatitude meteorology where surface temperature is approximately 10 °C cooler. The retrieved $P_{top}$ distribution shifts as expected with median $P_{top}$ bias increasing from 12 hPa to 15 hPa.

Overall the OCO-2 LUT gives better $P_{top}$ retrieval statistics than MODIS for these shallow marine clouds, where for these clouds MODIS retrievals rely on brightness temperature at $\lambda \sim 11$ μm and so may mis-assign $P_{top}$ when a temperature inversion is present (Baum et al., 2012). However, OCO-2 has a larger footprint, smaller swath and only retrieves during nadir view orbits. The $P_{top}$ bias relative to CALIPSO is concerning for a future optimal estimation retrieval, since biased prior properties may subsequently bias the posterior retrieved state in unpredictable ways. We confidently exclude aerosol

and meteorology as the main factors in the observed bias, and propose that the main candidate processes are a combination of horizontal variability, OCO-2-CALIPSO collocation error and potentially vertical cloud structure. In the future OE retrieval we would expect horizontally non-uniform clouds to produce spectra that are more difficult to match under our RT assumptions, so such cases may be identified by the posterior $\chi^2$ statistics. For vertical structure biases, we plan a detailed future investigation.

Retrieved $\tau$ and $r_e$ show good correlation with those from MODIS and the variance of the differences is smaller than implied by the MODIS-reported uncertainties, if the LUT and MODIS uncertainties are independent Gaussian with MODIS' reported variance. However, OCO-2's $I_{c,O_2}$ instrumental noise is lower than MODIS' (single channel signal-to-noise ratio, SNR of 300—1200, versus the MODIS band 4 specified SNR of 228), so the instrumental uncertainty contribution to the error budget should be smaller for OCO-2. There are also common characteristics between the retrievals, such as the use of

fixed droplet size distribution variances, so individual footprint error will covary between the two. Such covariance should



further reduce the inter-satellite difference in retrieved $\tau$ and $r_e$. A quantitative analysis would require a thorough calculation controlling for individual terms in the error budget, we simply conclude that there is no evidence of substantial unexpected variance in our retrieved $\tau$ and $r_e$.

Of greater concern is the residual OCO-2 minus MODIS differences of 0.77 in $\tau$ and to a lesser extent -0.25 µm in $r_e$. For $\tau$
the bias increases both with horizontal homogeneity and with $\tau$, and we expect to be able to identify these clouds scenes using retrieved $\tau$, and either OCO-2 developed metrics of spatial variability or future retrieval $\chi^2$.

For the $r_e$ bias we briefly assessed several factors. Horizontal variability tends to increase retrieved $r_e$ (Werner et al., 2018; Zhang et al., 2012) but we found no evidence of a strong dependence on spatial variability according to MODIS SPI. For a hypothetical calibration bias we ran the LUT retrieval with a -1 % shift in $I_{c,st}$, and this shifts median $r_e$ by +0.2 µm.
Instrumentally, the MODIS band 7 used in these $r_e$ retrievals begins at $\lambda = 2.105$ µm, outside the strong $CO_2$ band. Changes in $CO_2$, or, more likely, temperature and vapour-driven broadening or vapour absorption could affect retrieved $r_e$. When retrieving with the midlatitude profile LUT described above, the median retrieved $r_e$ increases by 0.17 µm. Given that the $r_e$ discrepancy is small, we make no further efforts to explain or reduce it.

## 5.   Summary and Conclusions

Here we developed a new pre-processor for a future optimal estimation retrieval using the OCO-2 A-band to provide new estimates of droplet number concentration in marine water clouds. This future retrieval aims to address limitations in the previously published OCO2CLD-LIDAR-AUX product, by (1) removing the requirement for collocated CALIPSO data now that the satellites are no longer formation flying and (2) adding OCO-2 information about $r_e$ to extend the analysis to droplet number concentration. The pre-processor must identify footprints that likely contain liquid clouds of sufficient $\tau$, and provide
prior properties for the future cloud retrieval. It may also be useful for identifying appropriate footprints on which other researchers could conduct partial column XCO2 retrievals.

The pre-processor first flags potentially cloudy scenes using a multi-layer perceptron network fed with continuum radiances across all 3 OCO-2 bands plus a set of absorption band radiances from the $O_2$ A-band. The next stage of the retrieval is to use a three-dimensional lookup table that that jointly retrieves $\tau$, $r_e$ and $P_{top}$ using radiances from two bands plus an A-band
absorption ratio. Footprints whose radiances are inconsistent with the lookup table, or whose implied $P_{top}$ occurs where $T < 0$ °C can also be excluded from future retrievals. These footprints were associated with worse fit statistics in OCO2CLD-LIDAR-AUX, implying that the new pre-processor will minimise the waste of computational resources on poor quality retrievals.

This pre-processor flag shows excellent agreement with MODIS and OCO-2, and the lookup table $\tau$ and $r_e$ compare well
with MODIS, while its $P_{top}$ shows better retrieval statistics than MODIS, when taking CALIPSO as the truth. Many of the



inter-satellite differences are associated with known factors: false positives from the classifier occur when scenes contain broken or multi-layered clouds, and the τ retrieval bias grows with the horizontal heterogeneity of the scene.

A main concern is that the median OCO-2 retrieved $P_{top}$ is closer to the surface than CALIPSO's, by approximately 12 hPa (~120 m). The assumed mean cloud extinction or its profile will affect photon paths lengths and so could introduce a bias in
retrieved $P_{top}$, and we propose that a detailed analysis of cloud vertical structure is the next and final step before the development of a new OCO-2 cloud retrieval. If successful, this new retrieval would add independent information on cloud droplet number concentration, allowing attempts to resolve apparent disagreements about low cloud processes.

**Code availability** sklearn is on github at: https://github.com/scikit-learn/scikit-learn and ReFRACTOR is at: https://github.com/ReFRACtor/framework


**Data availability** OCO2CLD-LIDAR-AUX can be downloaded from the CloudSat Data Processing Center at: http://www.cloudsat.cira.colostate.edu/data-products/level-aux/oco2cld-lidar-aux

**Supplementary Information** [attached .pdf]

**Author contributions** MR contributed to study design, ran the analyses and wrote the paper. JM set up ReFRACtor for cloudy scene simulations and provided technical support. MDL and GLS contributed to study design, analysis and paper drafting.

**Competing interests** The authors declare no competing interests.

**Acknowledgements** This work was carried out at the Jet Propulsion Laboratory, California Institute of Technology under a contract with the National Aeronautics and Space Administration. Government sponsorship acknowledged. MR gratefully acknowledges Dr. Aronne Merrelli for his updated cloud scattering property tables that corrected an error in $CO_2$ band
absorption and removed resultant biases in retrieved $r_e$. MR also thanks Annmarie Eldering and Michael Gunson for helpful input related to the OCO-2 mission.

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



**Table 1. Summary of methods for determining properties in OCO2CLD-LIDAR-AUX, and changes introduced in this study. OCO2CLD-LIDAR-AUX is a full optimal estimation (OE) retrieval that combined CALIPSO and OCO-2 information to obtain its prior state. This study is intended to provide OCO-2 only prior information for a future OE retrieval.**

| Property | OCO2CLD-LIDAR-AUX | This study |
|---|---|---|
| Cloud flagging | 1. CALIPSO single layer cloud<br>2. CALIPSO $P_{top} > 680$ hPa<br>3. OCO-2 radiances exceed static thresholds<br>4. OCO-2 weak/A-band radiance ratio above fixed threshold for given A-band radiance | Multi-layer perceptron network classification based on OCO-2 radiances |
| Cloud effective radius ($r_e$) | Fixed $r_e = 12$ μm in retrieval | Estimated from OCO-2 radiances via 3d lookup table |
| Cloud top pressure ($P_{top}$) | Prior from collocated CALIPSO 01kmCLay, posterior from OE retrieval | Estimated from OCO-2 radiances via 3d lookup table |
| Cloud optical depth (τ) | Prior from lookup table map to A-band radiance, posterior from OE retrieval | Estimated from OCO-2 radiances via 3d lookup table |
| Cloud geometric thickness ($H$) | Prior from subadiabatic model using prior, | Not reported: assumed subadiabatic where needed for radiative transfer, derived from τ, $P_{top}$, $r_e$. |
| Retrieved properties | τ, $P_{top}$, $H$ | τ, $P_{top}$, $r_e$ for prior in future retrieval. Prior $H$ and/or $N_d$ to be derived from subadiabatic model. |




**Table 2. Summary of datasets used. Note that we use the 01kmCLay and MYD061KM products collocated with OCO-2 as described in Taylor et al. (2016).**

| Dataset name | Long name | Summary of data used |
|---|---|---|
| L1bSc | OCO-2 Level 1b Science data | OCO-2 radiances, geographic information and geometry for radiative transfer, instrument information. |
| L2Met | OCO-2 Level 2 Meteorological data | Footprint T profiles for $T_{top}$, mean $T$ and $q$ profiles for LUT RT. |
| 01kmCLay | CALIPSO 1 km cloud layer product | Cloud layer presence, $P_{top}$ and phase from feature classification flag |
| MYD061KM | MODIS Aqua 1 km cloud product | Cloud effective radius and optical depth |
| OCO2CLD-LIDAR-AUX | OCO-2 CALIPSO combined cloud retrieval | Preprocessor throughput, retrieval $\chi^2$ and retrieval cloud properties |

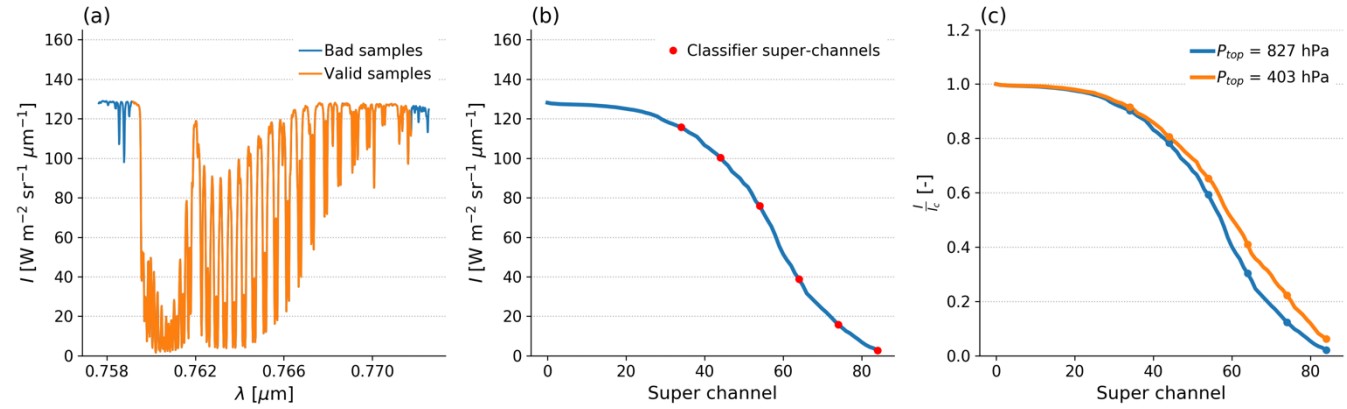


**Figure 1.(a) OCO-2 A-band spectrum for a cloud with CALIPSO $P_{top}$ = 827 hPa, with the used channels in orange and non-used (e.g. due to FPA pixel damage) in blue, (b) the smoothed super-channel spectrum, where the channels are ranked in brightness and then non-overlapping 10-channel means are taken. The super channels used in the classifier are shown in red. (c) Comparison of the ranked ratio $I/I_c$ between this cloud, and a higher altitude ($P_{top}$ = 403 hPa). The SZA is within 0.01° and the continuum**
**radiance within 1 % between each spectrum, the differences are largely due to the shorter path length resulting in less absorption for the higher altitude cloud.**





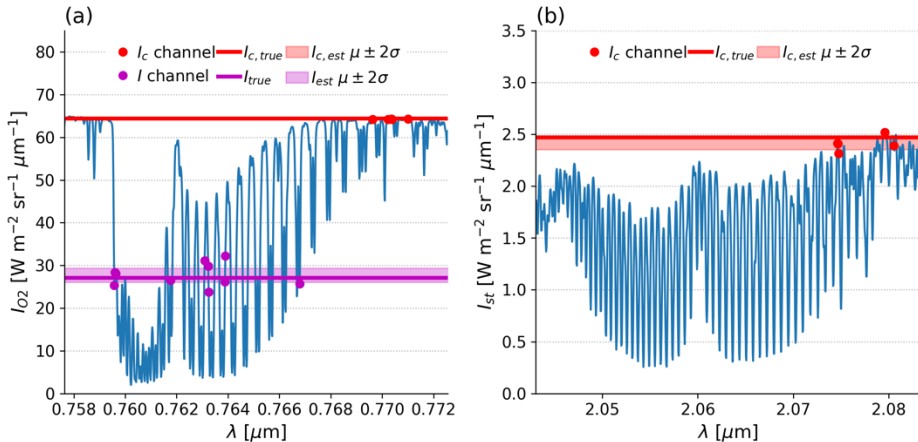

**Figure 2. Mean cloudy scene spectra in (a) the O$_2$ A-band and (b) the strong CO$_2$ band. The channels used in the lookup tables are shown as points, red for the continuum radiance $I_c$ and magenta for the O$_2$ absorption band radiance $I$. Thick horizontal lines represent the "truth", either the L1bSc file's continuum radiance for $I_c$ or the mean of the 600—609th brightest undamaged channels (i.e. the 60th super-channel) for $I$. The shaded bands of the same colour are the mean±2 standard errors based on the selected channel sample sizes (5 for $I_c$, 10 for $I$).**

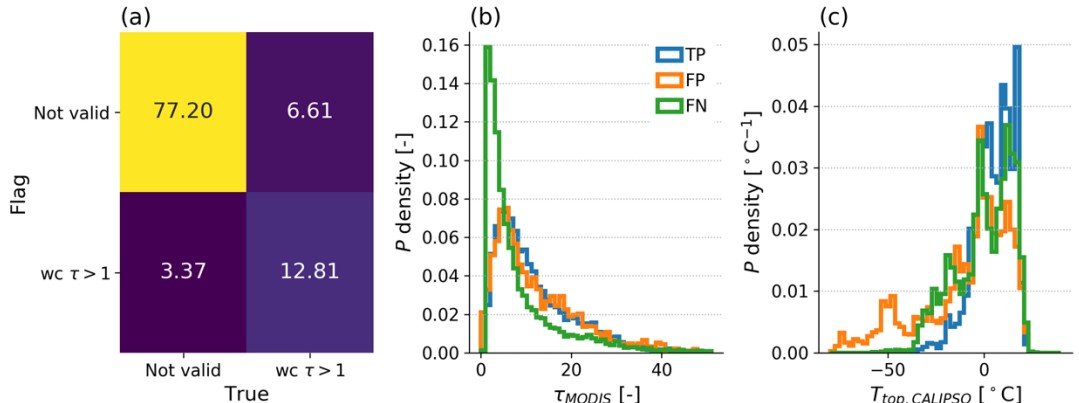

**Figure 3. (a) confusion matrix with values in %, comparing trained classifier ("flag") with collocated MODIS-CALIPSO definitions ("truth") and entries being classified as single-layer water cloud (wc) with $\tau > 1$, or "not valid", (b) normalised histograms of collocated MODIS $\tau$ where retrieved, for true positives (TP), false positives (FP) and false negatives (FN), (c) normalised histograms of collocated CALIPSO $T_{top}$ where retrieved, colours as in (b).**

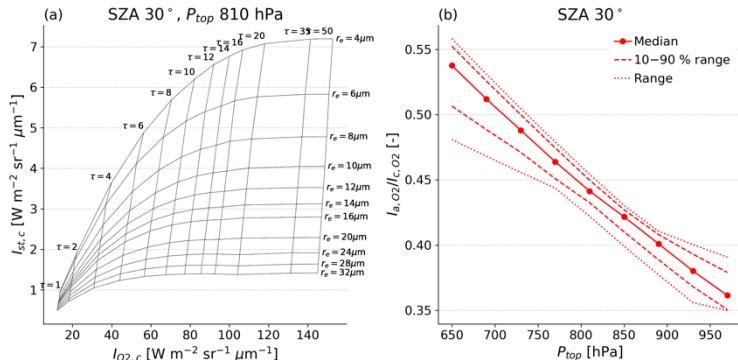

**Figure 4. Example lookup table (LUT) properties. (a) radiance in the strong CO$_2$ continuum as a function of A-band continuum**
**radiance at SZA = 30° for clouds with $P_{top}$ = 810 hPa and $\tau$, $r_e$ as labelled, (b) A-band absorption ratio within the SZA = 30° table**
**as a function of $P_{top}$. The solid line is the median value within each LUT at a fixed $P_{top}$, the dashed lines span the 10—90 % range**
**and the dotted lines span the minimum to maximum values.**

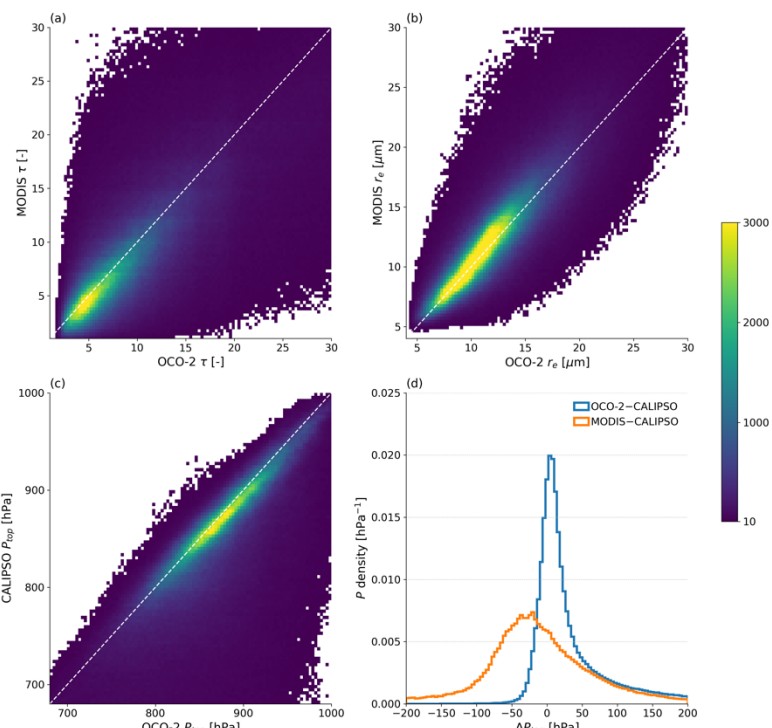

**Figure 5. Inter-satellite comparison of retrieved cloud properties, (a) MODIS vs. OCO-2 $\tau$, (b) MODIS vs. OCO-2 re, (c)**
**CALIPSO vs. OCO-2 $P_{top}$ and (d) OCO-2 or MODIS $P_{top}$ minus CALIPSO $P_{top}$. The colour bar on the right applies to (a)—(c).**

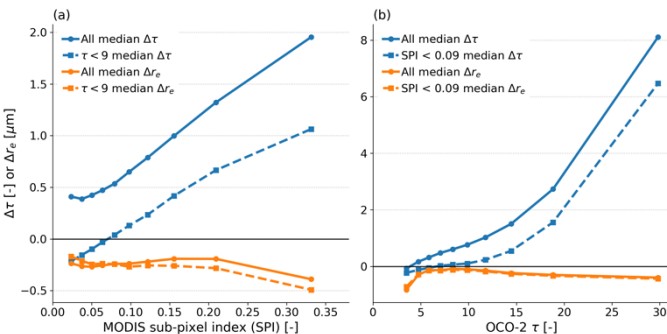

**Figure 6. Binned median bias in OCO-2 minus MODIS τ (blue) or $r_e$ (orange) when (a) binned by MODIS sub-pixel index (SPI)**
**derived from the 250 m sampling at λ = 0.66 μm or (b) binned by the OCO-2 LUT retrieved τ. Solid lines are for the full samples,**
**and dashed lines are for the subset (a) below the median OCO-2 τ or (b) below the median MODIS SPI.**

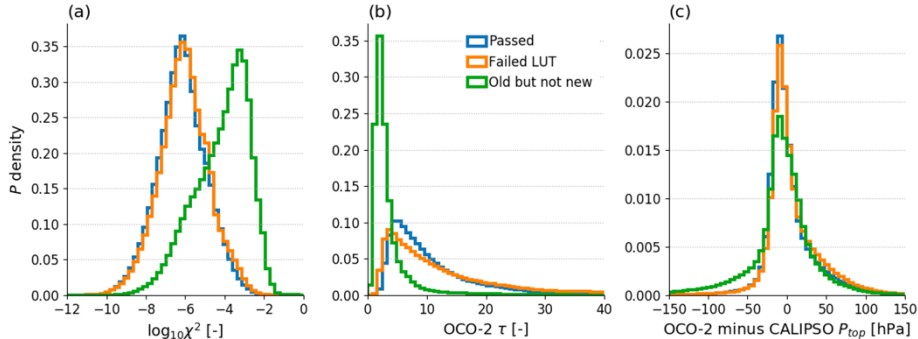

**Figure 7. Normalised histograms of OCO2CLD-LIDAR-AUX outputs where SZA < 45° separated into whether the soundings pass**
**the new pre-processor flag and retrieval or not. The "passed" set are those that returned valid cloud properties from the LUT**
**along with $T_{top} > 0$ °C, the "failed LUT" case gave invalid cloud values or had $T_{top} < 0$ °C, and the "old but not new" set are those**
**that were attempted in OCO2CLD-LIDAR-AUX but are not passed by the new classifier. (a) Logarithm of $\chi^2$, (b) retrieved τ, (c)**
**$P_{top}$ minus the closest CALIPSO retrieved $P_{top}$.**