# Peer review of "A new OCO-2 cloud flagging and rapid retrieval of marine boundary layer cloud properties"

_Atmospheric Measurement Techniques, 2020_

## Referee Comment (RC1) · Anonymous Referee #1 · 12 Jun 2020

The paper is well written, and in principle can be published as it is now.

I have two minor questions:

Line 156: I am confused with the sentence, that LIDORT assumes semi-infinite atmosphere. What is the benefit in assuming semi-infinite atmosphere? How do you determine the most meaningful wavelengths?
* * *

---

## Referee Comment (RC2) · Anonymous Referee #2 · 14 Jun 2020

**"A new OCO-2 cloud flagging and rapid retrieval of marine boundary layer cloud properties" by Mark Richardson[1,2], Matthew D. Lebsock[1], James McDuffie[1], Graeme L. Stephens**

The manuscript documents a new cloud flagging and retrieval pre-processor for OCO-2 marine boundary liquid cloud retrieval algorithm. This new pre-processor contains two steps: step one uses a machine learning perceptron network to classify low liquid cloud, and step two that simultaneously retrieves optical thickness (t ), effective radius and cloud top pressure (Ptot) with a three-parameter LUT using CO2, and O2 continnum radiances and A-band absorption to continuum ratio. The new retrievals and throughput are compared with MODIS and CLIPSO retrievals and original CLD-LIDAR-AUX as needed. In general, the new classifier/retrieval without CLIPSO show promising results. The residual biases in optical thickness, effective radius and cloud top pressure are discussed that could be potentially linked to spatial variability and in-cloud vertical structure. I would like to see how changes in pre-processor retrieval of tau, Re and Ptot affect the eventual OE retrieval beyond the impact of qualifying throughput. But that could be another major undertake, if the authors are planning to change the cloud model within the OE retrieval system and that could be content of the next paper.

The paper is well written and the addition of Table 1 is very helpful. I have a few more minor comments as follows:

P1L25-33. The description of the first paragraph is a bit misleading as if A-band alone allows all the retrieval, i.e., Tau, Ptop, H and Nd.  But A-band only helps with cloud flag and cloud top height, other channels are needed to retrieval Tau, Re, and H.

P3L83. Why do you limit to nadir only orbits?

Figure 3. A-band ratio will be less sensitive for low clouds. The 403 hPa cloud shown is way above the boundary clouds. More relevant to the retrieval of low clouds is whether you are able to separate clouds from 827 hPa to something like 750 km.

P7L221: Are you sure the middle parameter is *Ic,st/Ic,O2,*  not Ic,st? In Figure 4a, LUT shows Ic,o2 and Ic,st as controlling parameters.

Table 1. It is better to spell out that the input to machine learning classifier includes both radiances and radiance ratios.

P7L220. Not clear if you process all pixels with LUT, or only pixels that identified as cloud from classifier to further process with LUT. If so, please mention here.

P9L280. Could the asymmetric bias of Ptot be related to truncation of LUT, i.e., the same A-band ratio could occur in higher cloud but smaller optical depth? How will the retrieval change if you have LUT that covers more pressure levels, i.e., from 500hPa to 1010 hPa?

Figure 7. I would rename the "passed", "failed LUT","old but not new" into "passed both","passed classifier","failed both".

P13L380 Do you mean "inhomogeneity"?

P13L384. You have previously scaled the $Ic,st$ channel by 0.9804. Is this scaling sufficient? If you scale by a smaller value, the mean bias of Re might be reduced.

---

## Author Comment (AC1) · 12 Jul 2020

The paper is well written, and in principle can be published as it is now.I have two minor questions:

**Thank you for your positive review. We answer your specific questions below with our text in bold.**

**We tried to keep our text changes as short as possible to maintain the manuscript's flow. This response expands in more detail to provide context so you can confidently judge whether our streamlined text is clear & accurate.**

Line 156: I am confused with the sentence, that LIDORT assumes semi-infinite atmosphere. What is the benefit in assuming semi-infinite atmosphere?

**You're right to be confused, our phrasing was bad so we have changed it. The OCO-2 RT bolts together multiple codes. Multiple scattering is very important for us so we specifically mention LIDORT,[1] which uses an infinite-medium solution method for the radiative transfer equation particular integral. However, the OCO-2 code does not otherwise assume a semi-infinite atmosphere so we removed "semi infinite".**

**Single scattering is handled separately[2], then there's the second-order-of-scattering code[3] polarisation correction and also the low stream interpolator[4] (which is also 2OS corrected), based on a successive order of interaction[5] approach. ReFRACtor combines these as described in O'Dell et al. (2012)[6]. There are a lot of assumptions and caveats here, and many decisions and supporting evidence underlie the algorithm design. Reporting everything would be unwieldy, so we now point at O'Dell et al. (2012), from which interested readers can reconstruct the full methodology.**

**The new phrasing is:**
**"The forward RT simulations used to generate the LUTs are performed with the ReFRACtor RT code, which implements the methodology described in Section 2.2.4 of O'Dell et al. (2012). Of particular relevance for cloudy scenes, multiple-scattering is calculated using LIDORT with a polarisation correction for low orders of scattering (Natraj and Spurr, 2007; Spurr, 2006). This assumes a plane-parallel atmosphere with a correction to the direct beam to account for Earth's sphericity."**

**We have endeavoured to include all relevant information while maintaining brevity.**

1. **Spurr, R. & Christi, M. The LIDORT and VLIDORT Linearized Scalar and Vector Discrete Ordinate Radiative Transfer Models: Updates in the Last 10 Years. in *Springer Seeries in Light Scattering* (ed. Kokhanovsky, A. A.) 1–62 (Springer Nature, 2019). doi:10.1007/978-3-030-03445-0_1**
2. **Nakajima, T. & Tanaka, M. Algorithms for radiative intensity calculations in moderately thick atmospheres using a truncation approximation. *J. Quant. Spectrosc. Radiat. Transf.* 40, 51–69 (1988).**
3. **Natraj, V. & Spurr, R. J. D. A fast linearized pseudo-spherical two orders of scattering model to account for polarization in vertically inhomogeneous scattering–absorbing media. *J. Quant. Spectrosc. Radiat. Transf.* 107, 263–293 (2007).**
4. **O'Dell, C. W. Acceleration of multiple-scattering, hyperspectral radiative transfer calculations via low-streams interpolation. *J. Geophys. Res.* 115, D10206 (2010).**
5. **Heidinger, A. K., O'Dell, C., Bennartz, R. & Greenwald, T. The Successive-Order-of-Interaction Radiative Transfer Model. Part I: Model Development. *J. Appl. Meteorol. Climatol.* 45, 1388–1402 (2006).**
6. **O'dell, C. W. *et al.* The ACOS CO 2 retrieval algorithm – Part 1: Description and validation against synthetic observations. *Atmos. Meas. Tech.* 5, 99–121 (2012).**

How do you determine the most meaningful wavelengths?

**We think this is still referring to the text following L156 and specifically "Angular output is calculated for a handful of wavelengths". We have expanded on the description:**

**"Angular output is calculated with 8 streams for predefined bins in gas optical depth while single stream calculations are done for preselected wavenumbers at a mean separation $\Delta\nu \sim 0.04$ $cm^{-1}$, with smaller separation within absorption bands. The high- and low-stream outputs are combined using O'Dell (2010)'s low-stream interpolation to rapidly and accurately reproduce high-stream output at all wavenumbers. These are then interpolated onto a uniform $\Delta\nu=0.01$ $cm^{-1}$ grid and convolved with the instrument line shapes (ILS) to obtain channel radiances."**

**We used a standard configuration provided by the OCO-2 team based on their optimised sampling. We had wanted to keep all discussion in wavelength rather than wavenumber, but the code implements this part in wavenumber so we drop consistency for precision, i.e. "uniformly at $\Delta\nu=0.01$ $cm^{-1}$" over "close-to-uniformly at $\Delta\lambda \sim 0.00059$ nm".**

---

## Author Comment (AC2) · 12 Jul 2020

Firstly, thanks for having read this paper so attentively and providing a review that both catches minor mistakes and is thought provoking with regards to the main science and context. We genuinely appreciate your efforts and respond to each point in detail below. Our text continues in bold, we leave yours unbolded, and we insert the updated figures at the end of this document.

**"A new OCO-2 cloud flagging and rapid retrieval of marine boundary layer cloud properties" by Mark Richardson1,2, Matthew D. Lebsock1, James McDuffie1, Graeme L. Stephens**

The manuscript documents a new cloud flagging and retrieval pre-processor for OCO-2 marine boundary liquid cloud retrieval algorithm. This new pre-processor contains two steps: step one uses a machine learning perceptron network to classify low liquid cloud, and step two that simultaneously retrieves optical thickness (t), effective radius and cloud top pressure (Ptot) with a three-parameter LUT using CO2, and O2 continnum radiances and A-band absorption to continuum ratio. The new retrievals and throughput are compared with MODIS and CLIPSO retrievals and original CLD-LIDAR-AUX as needed. In general, the new classifier/retrieval without CLIPSO show promising results. The residual biases in optical thickness, effective radius and cloud top pressure are discussed that could be potentially linked to spatial variability and in-cloud vertical structure. I would like to see how changes in pre-processor retrieval of tau, Re and Ptot affect the eventual OE retrieval beyond the impact of qualifying throughput. But that could be another major undertake, if the authors are planning to change the cloud model within the OE retrieval system and that could be content of the next paper.

We are also excited to see the updated OE retrieval but want to do a comprehensive job. We first need to derive a correction for the cloud vertical structure bias we mention in this submission. After that we have to (i) recalculate the  $S_y$  covariance matrix accounting for vertical structure and given our new, smaller  $r_e$  uncertainty and (ii) repeat the channel selection algorithm of Richardson & Stephens (2018) to determine whether to continue using the 75 channels selected for OCO2CLD-LIDAR-AUX. We judge this to be too much content to add to this paper.

We expect the vertical structure issue to be important enough that doing more retrievals without accounting for it wouldn't be a good time investment.

The figure below shows (left)  $\tau$ =10 cloud simulated *I* and (right) changes in *I* when properties are the same but the vertical extinction structure is converted to subadiabatic (orange) or when the cloud is lowered by 10 hPa (blue). 10 hPa is multiple times our idealised OE posterior *P*top uncertainties (Richardson & Stephens, 2018, again), and the vertical structure effect on radiances is similar in magnitude to that, so we want to finish development of our correction method.

The paper is well written and the addition of Table 1 is very helpful. I have a few more minor comments as follows:

P1L25-33. The description of the first paragraph is a bit misleading as if A-band alone allows all the retrieval, i.e., Tau, Ptop, H and Nd. But A-band only helps with cloud flag and cloud top height, other channels are needed to retrieval Tau, Re, and H.

Agreed, leaving the  $r_e$  issue to the end of the paragraph makes it easy to misinterpret. We removed the last sentence and added:

", provided coincident information about effective radius  $(r_e)$  from other channels."

P3L83. Why do you limit to nadir only orbits?

**We have added the following justification:**

"...we use nadir only orbits to provide complementary vertical information on clouds that are too low or thin to be adequately profiled by CloudSat's nadir-view radar. Glint-view footprints would preclude our use of the nadir-only CALIPSO lidar data and atmospheric photon path lengths would be longer, thereby reducing the retrieval sensitivity. Given our retrieval's computational expense we limit to nadir orbits to optimise the likelihood of good retrievals."

Additional reasons include how this is largely a CloudSat project; the original CloudSat proposal included an A-band spectrometer for this purpose. We hope to use the  $\sim$ 5 years of coincident measurements by combining CloudSat's exquisite sensitivity to precipitation with an OCO-2  $N_d$  product plus ancillary information to do aerosol-cloud-precipitation science.

That said it would be great if we (or someone) had the time & resources to explore the potential to use the glint data.

Figure 3. A-band ratio will be less sensitive for low clouds. The 403 hPa cloud shown is way above the boundary clouds. More relevant to the retrieval of low clouds is whether you are able to separate clouds from 827 hPa to something like 750 km.

We assume this refers to Figure 1 and have changed to footprints containing clouds at 943 hPa and 740 hPa. This is a good suggestion to keep everything consistently focussed on low clouds.

We argue against showing smaller  $P_{top}$  gaps than this: this figure's purpose is to help readers to visualise spectrum differences due to changing photon path length. The ~200 hPa gap makes this visible. The rest of our analysis and past work shows sensitivities and error statistics. The new figure 1 and caption are at the end of this document.

P7L221: Are you sure the middle parameter is Ic,st/Ic,O2, not Ic,st? In Figure 4a, LUT shows Ic,o2 and Ic,st as controlling parameters.

We are absolutely not sure. That was a typo and has been corrected, thanks for your detailed reading.

This was a leftover from an earlier version when we used that ratio: it "stretches out" the vertical component of the Nakajima-King table so that instead of looking like a droopy slice of pizza it looks more like a wavy pizza al taglio.

The original interpolation code performed better with this approach at low  $\tau$ , but updates mostly removed the performance advantage so we decided to stick to the standard droopy pizza slice to avoid having to describe & justify an alternative.

Table 1. It is better to spell out that the input to machine learning classifier includes both radiances and radiance ratios.

**Agreed, text changed:**

**"...based on OCO-2 radiances and radiance ratios"**

P7L220. Not clear if you process all pixels with LUT, or only pixels that identified as cloud from classifier to further process with LUT. If so, please mention here.

**Added another step:**

**"(ii) Apply the classifier to identify appropriate cloudy footprints, pass only these to the next step"**

P9L280. Could the asymmetric bias of Ptot be related to truncation of LUT, i.e., the same Aband ratio could occur in higher cloud but smaller optical depth? How will the retrieval change if you have LUT that covers more pressure levels, i.e., from 500hPa to 1010 hPa?

We did not think about this but have now tested. Extending the LUT to 1,010 hPa would put cloud top below our surface pressure. We'd need to generate full new LUTs as well as reprocessing the retrievals. Our alternative is to cut down the current LUTs: we tried removing the 970 hPa entries, then removing the 650 hPa entries, then

removing both. The retrievals were then reprocessed. Note that we can still retrieve lower/higher values because our interpolator can "fill in"  $P_{top}$  values in some cases. Here's a 2d histogram of the retrieved  $P_{top}$  when the 970 hPa inputs are removed, versus our main results:

We first address the horizontal and vertical striping: it's where LUT  $P_{top}$  values are defined. We couldn't identify the exact reason. It could be due the Nakajima-King table "point" at low tau/reff where CO2-band radiances span a small range and could present interpolation problems. These "stripes" contain optically thin clouds (median  $\tau$ =3.5 versus population  $\tau$ =9.7) with small droplets (median  $r_e$ =5.8 µm versus population  $r_e$ =12.0 µm), i.e. they are in the point of the N-K pizza slice. This is <0.02 % of the sample so we don't consider it to be worth further development at this point.

Back to the rest of that 2d histogram: the top right cluster shows disagreement where LUT information has been removed. The interpolation method still generates some retrievals there, and gives high or low biases depending on the original  $P_{top}$ .

This suggests to us that the limitations placed on the LUT  $P_{top}$  values are not related to our  $P_{top}$  bias relative to CALIPSO. Referring to Figure 5(c), the large biases occur for  $P_{top}$  values well within the table rather than at the edges as we'd expect if it were related to our truncated table.

Here are the median and 10—90 % ranges of retrieval differences when we restrict the input LUT  $P_{top}$  values: the median bias ranges from -0.8 hPa to 0.1 hPa, versus our ~12 hPa median bias.

| LUT Ptops removed | Median [10%,90%] new minus full table P top |
|-------------------|--------------------------------------------------------|
| 970 hPa           | 0.07 [-1.42,2.29]                                      |
| 650 hPa           | -0.78 [-3.98,6.84]                                     |
| 650, 970 hPa      | 0.00 [-0.08,0.09]                                      |

Our LUT intentionally extends to 650 hPa, i.e. beyond the ISCCP "low cloud" threshold of 680 hPa, which was an attempt to minimise truncation issues. This was done even though we intend to focus on the lower altitude clouds. Given the apparently small effect we have not made any changes to the paper in response to this comment. Figure 7. I would rename the "passed", "failed LUT", "old but not new" into "passed both", "passed classifier", "failed both".

This is nicer and clearer. However, since we don't pass the non-classified cases to the LUT [now clarified in response to P7L220 comment] it feels a bit weird to say the final group "failed both". We reproduced the figure using "pass both", "passed classifier", "failed classifier" and changed the legend. We also increased label font size a bit. The figure and its caption are at the end of this document.

P13L380 Do you mean "inhomogeneity"?

**Another typo, changed.**

P13L384. You have previously scaled the Ic,st channel by 0.9804. Is this scaling sufficient? If you scale by a smaller value, the mean bias of Re might be reduced.

This is true and it's unfair to apparently "accuse" calibration when it could be due to one of our choices. We had thought about this but attentive readers should definitely question this, so now we provide our justification by replacing that sentence with:

"We also ran the LUT retrieval with a -1 % shift in  $I_{c,st}$ , which shifts median  $r_e$  by +0.2  $\mu$ m. Such a radiance shift could be necessary due to errors in calibration or in our derived scaling factor of 0.9807, which we used to relate the L1bSc file  $I_{c,st}$  to our lookup table channels. We could therefore reduce our  $r_e$  bias by further scaling the  $I_{c,st}$  radiances, but the scaling was derived from directly comparing the channel radiances rather than as a *post hoc* correction to improve retrieval results. If the  $r_e$  bias is due to other factors then this *post hoc* correction could result in compensating errors which hide other flaws in the retrieval."

Figure 1.(a) OCO-2 A-band spectrum for a cloud with CALIPSO  $P_{top} = 943$  hPa, with the used channels in orange and non-used (e.g. due to FPA pixel damage) in blue, (b) the smoothed super-channel spectrum, where the channels are ranked in brightness and then non-overlapping 10-channel means are taken. The super channels used in the classifier are shown in red. (c) Comparison of the ranked ratio  $I/I_c$  between this cloud, and one at higher altitude ( $P_{top} = 740$  hPa). The SZA is within 0.01° and the continuum radiance within 1 % between each spectrum, the differences are largely due to the shorter path length resulting in less absorption for the higher altitude cloud.